# Extinction risk modeling predicts range-wide differences of climate change impact on Karner blue butterfly (*Lycaeides melissa samuelis*)

Yudi Li[1]*, David Wilson[2], Ralph Grundel[3], Steven Campbell[4‡], Joseph Knight[5‡], Jim Perry[6‡], Jessica J. Hellmann[7]

1 Energy Graduate Group, University of California Davis, Davis, CA, United States of America, 2 Minnesota Department of Natural Resources, Grand Rapids, MN, United States of America, 3 US Geological Survey, Lake Michigan Ecological Research Station, Chesterton, IN, United States of America, 4 Albany Pine Bush Preserve Commission, Albany Pine Bush, NY, United States of America, 5 Department of Forest Resources, University of Minnesota, St. Paul, MN, United States of America, 6 Department of Fisheries, Wildlife and Conservation Biology University of Minnesota, St. Paul, MN, United States of America, 7 Conservation Sciences Graduate Program, University of Minnesota, St. Paul, MN, United States of America

☯ These authors contributed equally to this work.
‡ These authors also contributed equally to this work
* evoli@ucdavis.edu

**Data Availability Statement:** The coordinates of 48 permanent sampling sites, density-dependent

## Abstract

The Karner blue butterfly (*Lycaeides melissa samuelis*, or Kbb), a federally endangered species under the U.S. Endangered Species Act in decline due to habitat loss, can be further threatened by climate change. Evaluating how climate shapes the population trend of the Kbb can help in the development of adaptive management plans. Current demographic models for the Kbb incorporate in either a density-dependent or density-independent manner. We instead created mixed density-dependent and -independent (hereafter "endo-exogenous") models for Kbbs based on long-term count data of five isolated populations in the upper Midwest, United States during two flight periods (May to June and July to August) to understand how the growth rates were related to previous population densities and abiotic environmental conditions, including various macro- and micro-climatic variables. Our endo-exogenous extinction risk models showed that both density-dependent and -independent components were vital drivers of the historical population trends. However, climate change impacts were not always detrimental to Kbbs. Despite the decrease of population growth rate with higher overwinter temperatures and spring precipitations in the first generation, the growth rate increased with higher summer temperatures and precipitations in the second generation. We concluded that finer spatiotemporally scaled models could be more rewarding in guiding the decision-making process of Kbb restoration under climate change.

(derived from count) and -independent data to run the models, and the sample R codes of modeling are contained in Supporting Information files (S1, S2, and S3, respectively). The raw population count data are not contained in this submission because they were shared to us by other Karner blue scientists under constrained permissions. Researchers interested in these original data are encouraged to contact: Indiana Dunes National Park Population Data (1994-2011) - Randy Knutson (US National Park Service), email: Randy_Knutson@nps.gov Central & Northwestern Wisconsin Population Data (1990-2018) - Ann & Scott Swengel, email: aswengel@jvlnet.com Fort McCoy, WI Population Data (1997-2018) - Tim Wilder (Department of Defense, Fort McCoy), email: tim timothy.t.wilder.civ@mail.mil Albany Pine Bush Population Data (1995-2018) - Steven Campbell (Albany Pine Bush Preserve Commission), email: SCampbell@albanypinebush.org.

**Funding:** The author(s) received no specific funding for this work.

**Competing interests:** The authors have declared that no competing interests exist.

## Introduction

Butterflies are benchmark indicator species due to acute sensitivities to abiotic environments [1–6]. Climate change could interfere with their lifecycle developments and interactions with host plants [7,8]. The resulting responses of butterflies in terms of survival, morphology, phenology, reproduction, and geographical distribution under novel conditions are well documented, potentially jeopardizing species that have reached the limit of adaptive capacity [9–12]. However, the impacts are mixed: it was found that warmer temperatures decreased larva survival and adult body size and concurrently enhanced egg and pupa survival, advanced flight period, altered female fertility, and induced extra generations [13–18]. Furthermore, drought can reduce food availability and delay growth, but excessive rainfall may disrupt egg laying and foraging of nectar and pollen [19–22].

The Karner blue butterfly (*Lycaeides melissa samuelis*, or Kbb), first described by Vladimir Nabokov in 1944, was listed as federally endangered in 1992 [23–25]. Like many species in the family Lycaenidae, it is vulnerable to climate change due to limited dispersal ability (< 1 km), single host plant (i.e., monophagy on wild lupine, *Lupinus perennis*), and habitat specialist of high-quality oak savanna and pine barren which have been lost and fragmented for decades as the result of fire suppression, agricultural intensification, and urbanization [26–30]. The Kbb is also experiencing new climate patterns, such as milder winter temperatures and prolonged summer droughts, further accelerating population declines [31–34]. Today, native Kbb populations are small and isolated within the United States and Canada in the states of New York, Michigan, and Wisconsin, extirpated in the states of Illinois, Indiana, and the province of Ontario, and reintroduced in the states of Ohio, New Hampshire, and Minnesota [35,36].

As a bivoltine species, first-generation Kbbs overwinter as eggs until early April, hatch as larvae feeding on wild lupines, become pupae undergoing a metamorphosis in mid-May, and emerge as adults in June with a lifespan of 5–7 days to reproduce. The second generation occurs during the summer, with hatching, pupation, and emergence in early June, late June, and early July, respectively [37]. Their thermal and drought tolerances are specific to each life stage [38–41]. Studies have shown that climatic stresses experienced by one stage influence the next by the 'carry-over effect,' and stresses experienced in previous generations also affect subsequent generations by the 'transgenerational effect' [42–45]. Therefore, attempts to understand climate impacts on Kbb require the consideration of both carry-over and transgenerational effects throughout the lifecycle [46–50].

Microclimates are usually decoupled from broad climatic gradients and are mainly determined by terrain and canopy coverage [51–54]. For instance, south-facing slopes are generally warmer than north-facing slopes in the northern hemisphere; higher elevations are colder and drier than lower altitudes [55,56]. Moreover, Fuller [57] ranked habitat loss, fragmentation, and weather-related loss of eggs/larvae as the top three factors contributing to Kbb population crashes. Strong density dependencies (i.e., per capita growth rate changes with density) were also nested within modeled demographic trends. These findings implicated that density-dependent and fine-scale, topo-climatic variables were critical to predicting seasonal and annual Kbb population variations [58–60].

In this study, we built extinction risk models for each of the two generations of five Kbb populations in the US. The models were endo-exogenous, capturing both density-dependent and density-independent factors that may interact in complex ways to regulate population growth rate instead of presence/absence typically utilized in species distribution models [61–63]. We hypothesized that i). higher temperature and precipitation could inhibit and promote the population growth of Kbbs, respectively, and ii). best-fitted models should vary across populations owing to their unique adaptations to local environments.

## Materials and methods

### Density-dependent data

We obtained access to the data for five Kbb populations across the species' range (Table 1). Four populations inhabited oak savannas in the Midwest U.S.: central Wisconsin (CW), north-western Wisconsin (NW), Fort McCoy (FM), and Indiana Dunes National Park (IDNP). One population inhabited the pine barrens in the eastern U.S.—Albany Pine Bush Preserve (APBP). The data represented adult Kbbs surveyed at multiple permanent sites (S1 Table) along one to several linear transects repeatedly surveyed over a 7–14 day interval during each of the two flight periods (i.e., from late May to late June for the first generation, and from mid-July to mid-August for the second generation) over 8–27 years.

As the number of surveys per flight period and the total length of transect per survey varied widely across researchers and years, we calculated mean counts per kilometer of transect generation-by-generation for each sampled site. We estimated density as the number of adult individuals per hectare (ha) by assuming that a kilometer of transect represented a 2.5-ha area [64]. We further calculated the population growth rate (λ) based on density change between the current and the previous generations.

### Density-independent data

We downloaded monthly climatic data from the Parameter-elevation Relationships on Independent Slopes Model (PRISM Climate Group) as raster maps at a resolution of 30 arc seconds (1/40th of a decimal degree). Each pixel was 800 m x 800 m [65,66]. PRISM's temperature and precipitation are gridded interpolations of climate data and digital elevation models (DEM) [67,68]. We examined four monthly climate variables–minimum temperature (IT), maximum temperature (AT), mean temperature (MT), and total precipitation (PT)–for each site and year surveys were conducted. As it is crucial to understand how climate interacts with Kbb life-cycles, we approximated the occurrence periods of three life stages for both generations by calculating the mean value of multiple consecutive months: egg (Dec-Mar as overwinter), larvae

**Table 1. Summary of flight-period counts of five Karner blue butterfly (*Lycaeides melissa samuelis*) populations.** The state, county, years, number of sites, total transect length per survey on average, total observed numbers of individuals throughout the project lifespan, total number of surveys throughout the project lifespan, and data sources are included. Refer to Fig 1 for the mapped locations of these five populations.

| Population | State | County | Year | Site | Transect Length (m) | Observed Individual | Survey Number | Data Source |
|---|---|---|---|---|---|---|---|---|
| Central WI[a] (CW) | WI | Jackson | 1990–2018 | 8 | 700 | 17514 | 892 | Ann & Scott Swengel (Independent Researchers) |
| | | Wood | 1990–2018 | 5 | | | | |
| | | Portage | 2000–2018 | 1 | | | | |
| NW[b] WI (NW) | | Burnett | 2005–2013 | 8 | 700 | 2442 | 576 | |
| Fort McCoy (FM) | | Monroe | 1997–2018 | 12 | 1049 | 23999 | 1595 | Tim Wilder (Department of Defense, Fort McCoy) |
| IN[a] Dunes National Park (IDNP) | IN | Porter | 1994–2011 | 6 | 800 | 30491 | 664 | Randy Knutson (IDNP) |
| Albany Pine Bush Preserve (APBP) | NY[a] | Albany | 2007–2018 | 8 | 980 | 132093 | 1049 | Steven Campbell (APB Commission) |

[a]WI: Wisconsin; IN: Indiana; NY: New York.

[b]NW: Northwest.

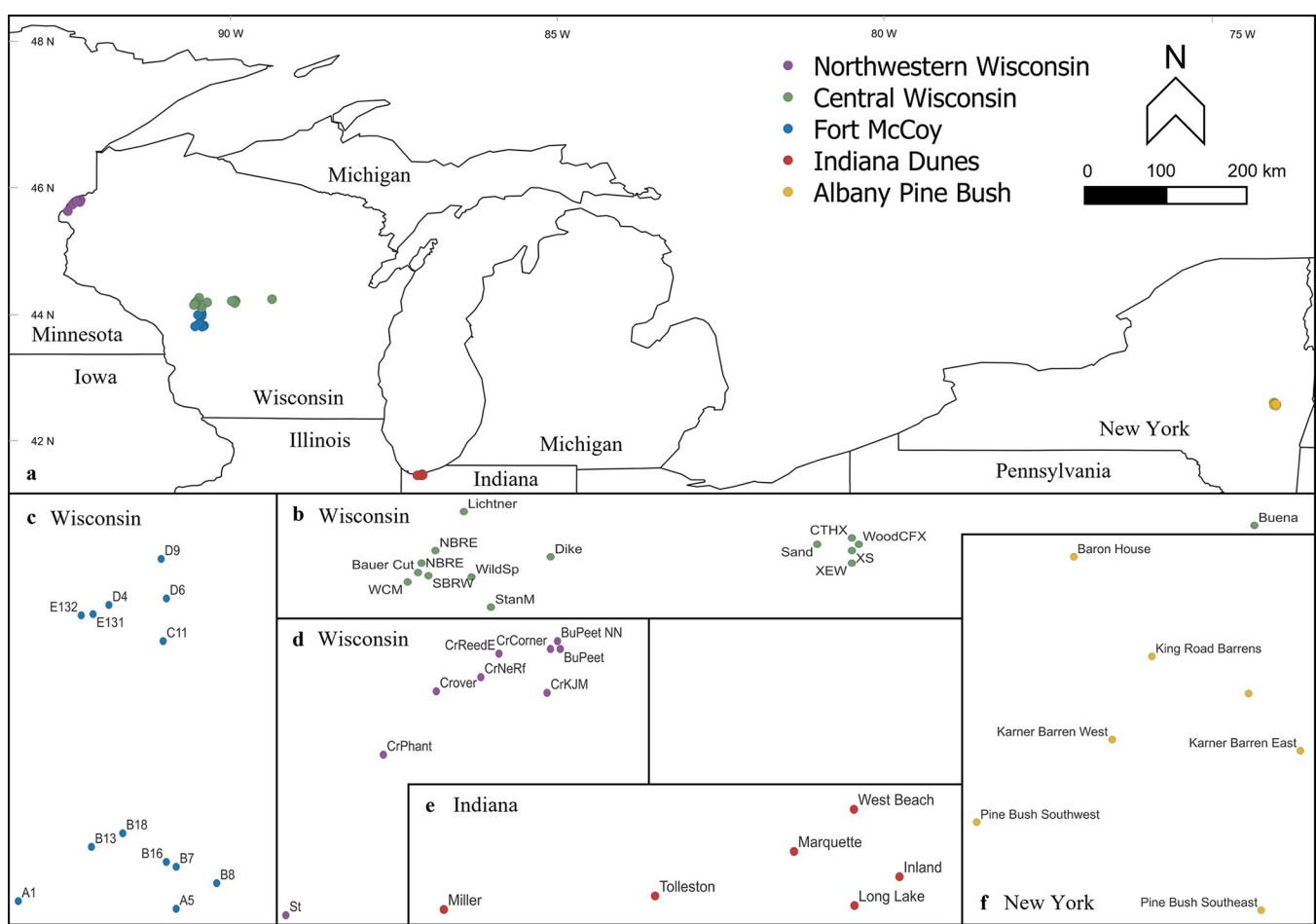

**Fig 1. Mapping of the 48 sampled permanent sites of five Karner blue butterfly (*Lycaeides melissa samuelis*) populations.** (a) Locations of all the sites. (b) Fourteen sites for populations in central Wisconsin. (c) Twelve sites for populations in Fort McCoy, Wisconsin. (d) Eight sites for populations in northwestern Wisconsin. (e) Six sites for populations in Indiana Dunes National Park, Indiana. (f) Eight sites for populations in Albany Pine Bush Preserve, New York.

and pupae (Apr-May as spring), and flight (Jun) for the first generation; egg (Jun), larvae and pupae (Jul), and flight (Aug) for the second generation.

We extracted three topographic variables–elevation, slope, and aspect—from 30 m U.S. Geological Survey (USGS) DEMs (in ArcGIS) [69]. The circular aspect layer was converted to a topographic solar radiation index (i.e., trasp) using the following linear transformation:

$$trasp = {}^{1} - \cos\left[\frac{\pi}{180}*(aspect - 30)\right]\Big/_{2}$$

The transformed output was a continuous variable between 0 and 1, where north-oriented slopes were assigned 0, and south-oriented slopes were 1 [70]. We also extracted canopy coverage from the 2011 National Land Cover Database (NLCD) and U.S. Forest Service (USFS) Tree Canopy Cover at 30 m resolution [71]. We assumed that topography and tree canopy cover had been constant over the past two decades [72].

## Statistical analysis

We included both density-dependent and -independent variables to build endo-exogenous models specific to (i) each population and (ii) all the surveyed locations of the five populations

of Kbbs pooled together (All). We modeled the first and the second generations separately because they were mostly non-overlapped except in June and experienced different climate patterns [73,74]. In total, we created 12 models: 6 populations * 2 generations. Each model had λ as the response variable and 18 standardized explanatory variables: two previous-generation densities, four (mean, minimum, and maximum temperatures, and total precipitation) * three (egg, larvae/pupae, and flight stages) monthly climates, three topographies (elevation, slope, and trasp), and tree canopy. As incorporating more than two previous generations was unlikely to increase explanatory power [65], we included densities of the first and the second generations in the previous year (Previous First (PF) and Previous Second (PS), respectively) for the first generation, and the densities of the second generation in the previous year (PS) and the first generation in the current year (CF) for the second generation. Due to various sampled sites across the five populations, we minimized estimation bias for All models using bootstrapping to balance oversampled and underrepresented populations [75,76]. The model equation is expressed as:

$$\lambda_t = \beta_0 + \beta_1 Y_{t-1} + \beta_2 Y_{t-2} + \beta_3 X1_t + \cdots + \beta_{17} X16_t + \varepsilon_t$$

where $\lambda_t$ is the density growth rate from the previous generation to the current generation; $Y_{t-1}$ and $Y_{t-2}$ are the densities of one and two generations ago, respectively; $Xn_t$ are the 16 macro- and microclimatic environmental factors in the current year. Since no "previous generations" existed for the first-year data, we started the models from the second year.

We applied random forest (RF, "randomForest" R package [77]) for regression because previous studies found that it was more robust to model population trends than regression-based or other machine-learning algorithms [78,79]. We then used the genetic algorithm (GA) in package "caret" for feature selection [80]. GA, as a computationally efficient approach, generated an entire population of feasible solutions that were repeatedly subjected to "cross-over" and "random mutation" until the combinatorial predictor optimum of the lowest Akaike Information Criterion (AIC) was identified [81]. However, GA risks overfitting, and RF output interpretation is often not straightforward. We, therefore, further applied jack-knife partial least squares (PLS) using the package "mdatools" [82] to identify the estimated coefficient, standard error, and p-value of each variable left in the "best" models. This linear latent approach is appropriate when the matrix of predictors has more variables than observations, especially with multicollinearities and missing values [83]. As a non-parametric approach, both RF and PLS make no assumptions about data distribution. The sample R codes were included in the S1 File.

## Ethics statement

The study did not involve human subjects, vertebrates, or cephalopods, nor any kind of animal sacrifice. It did involve the observation of a federally listed endangered species–the Karner blue butterfly. Approvals from the U.S. Fish and Wildlife Service to conduct Kbb counts were obtained on federal, state, and non-governmental organization lands managed by the U.S. National Park Service (Indiana Dunes National Park population), U.S. Department of Defense (Fort McCoy population), and the Albany Pine Bush Commission (Albany Pine Bush population), and on public-access land (state- and county-owned, or public access rights-of-way immediately along public roads–central and northwest Wisconsin populations). According to data owners, all the surveys were legally carried out without intentionally handling, harassing, or taking specimens. For our study, no permits were required to share the raw data with us for modeling and analysis.

## Results

In the first generation (Table 2), population growth rate λ decreased with increased density of previous-year second generation for the All model and the three populations in Wisconsin (central Wisconsin, northwestern Wisconsin, and Fort McCoy) with p-values < 0.05. However, for those in Indiana Dunes National Park and Albany Pine Bush Preserve, the density of previous-year first generation displayed the significant inverse correlation with λ (p < 0.01). λ was consistently smaller with higher overwinter temperatures (mean, max, and min), spring total precipitation, and June min temperature, though these negative relationships were not always significant. Overwinter precipitation and spring max temperature were only significant to the population in Albany Pine Bush Preserve (p < 0.001 and p < 0.01, respectively)–the only population not in the Upper Midwest. The rest of the macroclimate variables, including spring min temperature, June mean and max temperatures, and June total precipitation, were not contained in any best-fitted model. The GA approach selected none of the macroclimatic variables the model of northwestern Wisconsin–the population at the northern edge of Kbb's range. Despite being included in the final northwestern Wisconsin and Indiana Dunes National Park models, the four topo-climatic variables were nonsignificant (p > 0.05) for both populations.

**Table 2. Best-fit partial least-square (PLS) models for all the five populations of Karner blue butterfly (*Lycaeides melissa samuelis*) pooled and for each population during the first generation.** The adjusted-$R^2$ value of each model is added to the last row of the table.

| | All[d] | | CW | | NW | | FM | | IDNP | | APBP | |
|---|---|---|---|---|---|---|---|---|---|---|---|---|
| PS[e] | -0.80±0.30[a] | *[b] | -0.88±0.359 | * | -0.79±0.351 | * | -0.18±0.123 | * | -0.82±0.386 | ns | -0.46±0.218 | ns |
| PF | [c] | | 0.63±0.218 | * | | | | | -0.88±0.146 | ** | -0.68±0.120 | ** |
| MT_OW | -0.69±0.222 | * | | | | | -0.07±0.029 | ns | -0.10±0.016 | ** | -0.09±0.028 | * |
| MT_SP | 0.26±0.080 | ns | | | | | | | | | | |
| MT_JN | | | | | | | | | | | | |
| AT_OW | | | -0.98±0.252 | * | | | -0.03±0.027 | * | | | | |
| AT_SP | | | | | | | | | | | 0.22±0.041 | ** |
| AT_JN | | | | | | | | | | | | |
| IT_OW | -0.63±0.209 | * | -0.80±0.162 | ** | | | -0.11±0.038 | * | | | -0.19±0.070 | * |
| IT_SP | | | | | | | | | | | | |
| IT_JN | -0.32±0.139 | ns | -0.56±0.131 | * | | | -0.11±0.044 | * | | | | |
| PT_OW | | | | | | | | | | | -0.27±0.020 | *** |
| PT_SP | -0.53±0.053 | ** | -0.96±0.041 | *** | | | -0.16±0.049 | * | -0.28±0.170 | ns | | |
| PT_JN | | | | | | | | | | | | |
| Canopy | | | | | -0.05±0.062 | ns | | | -0.03±0.022 | ns | | |
| Elevation | | | | | -0.12±0.064 | ns | | | 0.08±0.072 | ns | | |
| Slope | | | | | -0.05±0.038 | ns | | | 0.11±0.070 | ns | | |
| Trasp | | | | | -0.02±0.106 | ns | | | 0.15±0.070 | ns | | |
| Adj-$R^2$ | 0.45 | | 0.43 | | 0.78 | | 0.46 | | 0.89 | | 0.62 | |

[a] Estimated coefficient ± standard error.

[b] *: p-value < 0.05

**: p-value < 0.01

***: p-value < 0.001; ns: p-value > 0.05.

[c] Blank cells represent that the variable was removed from the model after feature selection by genetic algorithm (GA).

[d] All: All five populations; CW: Central Wisconsin; NW: Northwestern Wisconsin; INDP: Indiana Dunes National Park; APBP: Albany Pine Bush Preserve.

[e] PS: Density of previous-year second generation; PF: Density of previous-year first generation; MT: Mean Temperature; AT: Maximum Temperature; IT: Minimum Temperature; PT: Total Precipitation; OW: Overwinter; SP: Spring; JN: June.

**Table 3. Best-fit partial least-square (PLS) models for all the five populations of Karner blue butterfly (*Lycaeides melissa samuelis*) pooled and for each population during the second generation.** The adjusted-$R^2$ value of each model is added to the last row of the table.

| | All[d] | | CW | | NW | | FM | | IDNP | | APBP | |
|---|---|---|---|---|---|---|---|---|---|---|---|---|
| CF[e] | -0.36±0.22[a] | ns[b] | -0.47±0.109 | ns | | | -0.92±0.537 | ns | | | 0.14±0.122 | ns |
| PS | 0.21±0.055 | * | 0.15±0.038 | * | 0.07±0.031 | ns | 0.40±0.152 | * | 0.08±0.069 | ns | 0.13±0.070 | * |
| MT_JN | [c] | | 0.05±0.017 | * | | | | | | | | |
| MT_JL | | | | | | | 0.05±0.031 | ns | | | | |
| MT_AG | | | | | 0.04±0.017 | * | | | | | 0.08±0.027 | * |
| AT_JN | | | | | | | | | | | | |
| AT_JL | 0.04±0.036 | * | 0.09±0.040 | ns | | | | | | | | |
| AT_AG | 0.07±0.037 | * | 0.09±0.038 | ns | 0.04±0.016 | ns | 0.14±0.078 | ns | | | 0.08±0.026 | * |
| IT_JN | 0.08±0.018 | * | | | | | 0.46±0.359 | ns | | | 0.08±0.030 | ns |
| IT_JL | | | 0.06±0.028 | ns | | | | | | | | |
| IT_AG | | | | | 0.07±0.021 | ns | | | | | 0.08±0.028 | * |
| PT_JN | | | | | | | | | | | | |
| PT_JL | 0.09±0.038 | ns | 0.09±0.024 | * | 0.05±0.033 | ns | 0.20±0.080 | * | 0.02±0.018 | ns | | |
| PT_AG | | | | | 0.04±0.032 | * | | | | | | |
| Canopy | | | | | 0.08±0.023 | * | | | 0.08±0.013 | * | | |
| Elevation | | | | | | | | | -0.03±0.017 | ns | | |
| Slope | -0.10±0.029 | * | -0.11±0.026 | * | | | -0.29±0.054 | ** | -0.09±0.019 | ** | 0.06±0.038 | ns |
| Trasp | | | | | | | | | -0.07±0.015 | * | | |
| Adj-$R^2$ | 0.21 | | 0.26 | | 0.31 | | 0.41 | | 0.46 | | 0.20 | |

[a] Estimated coefficient ± standard error.

[b] *: p-value < 0.05

**: p-value < 0.01; ns: p-value > 0.05.

[c] Blank cells represent that the variable was removed from the model after feature selection by genetic algorithm (GA).

[d] All: All five populations; CW: Central Wisconsin; NW: Northwestern Wisconsin; INDP: Indiana Dunes National Park; APBP: Albany Pine Bush Preserve.

[e] PS: Density of previous-year second generation; CF: Density of current-year first generation; MT: Mean Temperature; AT: Maximum Temperature; IT: Minimum Temperature; PT: Total Precipitation; JN: June; JL: July; AG: August.

In the second generation (Table 3), λ increased with higher density of previous-year second generation, June min temperature, July total precipitation, and August mean, min, and max temperatures in more than one population with p-values < 0.05). The λ also uniquely increased with higher June mean temperature and August total precipitation (p < 0.05) in central Wisconsin and northwestern Wisconsin, respectively. The density of the current-year first generation was never significant, and both June max temperature and total precipitation were absent in all models. Unlike the first generation, tree canopy and slope became significant in multiple populations with consistent positive and negative correlations, respectively. In particular, the p-values of slope were smaller than 0.01 in Fort McCoy and Indiana Dunes National Park–both populations were closer to the southern edge of Kbb's range than the other three populations. For those in Indiana Dunes National Park specifically, λ further decreased with larger trasp, and none of the density-dependent and macroclimatic variables were significant.

## Discussion

Best-fitted models predicting a measure of Kbb growth rate differed between generations and among the five populations we studied. However, the density of the previous-year second generation (PS) almost always had larger regression coefficients than any individual density-dependent variable. The association between population growth rate λ and PS was negative in

the first generation and positive in the second, verifying the findings for other multivoltine species that the population size of the second generation is typically three to four times larger than the size of the first generation from in the same year [84–86]. This pattern of two-generation lagging might be pertinent to the phenology of the larval hostplant, wild lupine: first-generation larvae feed on small, newly sprouted leaves in early spring, while they hatch approximately a week after eggs are laid on more mature leaves with, perhaps, higher nutrient quality during the second generation [87,88]. However, Grundel [89] found that under ordinary summer conditions, such as drought, leaf nutrients were often lower during the second generation compared to freshly emerging lupine in spring. Thus, an alternative mechanism could be that a higher overwinter mortality of eggs led to a lower density of emerging adults in late spring.

In the first generation of our studied five populations, the decrease of λ along with higher overwinter temperatures (i.e., mean, minimum, and maximum) and precipitation were congruent with previous studies on the butterflies [90–93]. There are three potential explanations for these population declines [57]: first, higher temperature accelerates the thawing of snow covers, resulting in quicker heat loss of eggs [94]; second, warm, moist conditions may cue earlier hatching from eggs in late winter when food sources are limited and environmental conditions are harsh [95–97]; third, butterfly eggs would be more vulnerable to diseases and fungal infections with higher temperature and precipitation [98]. In spring, precipitation was a better predictor of λ for Kbbs, unlike some other butterfly species for which higher temperature drove earlier maturation from the pupa and thus smaller body sizes and reduced fitness [99–108]. The negative association between λ and precipitation indicated that rainfall shortage might not necessarily hamper larval and pupal development, especially as the host wild lupine is a drought-tolerant native species [109]. The absence of June variables in models suggested that Kbbs may become much less sensitive to abiotic conditions after the emergence of adults.

In the second generation of these five populations, associations between λ and temperatures were consistently positive. A warmer summer was considered conducive to multivoltine species [110]. Higher June and July temperatures might enhance host plant quality and advance adult emergence, resulting in a higher chance of the appearance of an extra generation [85]. However, the envisaged third generation, if present, is worth further research as this could be a maladaptive response if novel conditions in the fall were unfavorable [111]. Lower August temperatures, especially the lower August maximum temperature, could trigger localized extinction, whereas a warmer August may promote egg laying due to greater nectar/pollen availability during extended flight periods [112,113]; hence, the benefit of warming could be passed to the next overwinter generation [114,115]. On the other hand, we detected uniformly positive associations of λ with July and August precipitations, suggesting that rainfall may alleviate water stress and shorten larval diapause [116,117]. In recent decades, the extinction of Kbb populations at Indiana Dunes National Park has, in part, been ascribed to prolonged summer droughts that led to extended diapause and host-plant desiccation, impairing their reproductivity [108,118].

Habitat specialist species like Kbbs can have a wider margin of thermal tolerance at the northern edge than at the southern edge of their range [3,61,119,120]. The asymmetrical response to thermal gradients [121] could explain why none of the macroclimatic variables were significant to northwestern Wisconsin populations during the first generation: theses populations were close to the northern edge of the range where annual temperatures were the lowest and growing seasons were the shortest. Under these conditions, lupine quality and phenology could be the limiting factors [122,123]. In contrast, Indiana Dunes National Park populations inhabited the southmost edge of the range and thus, were more likely to be exposed to intolerant weather conditions. Nevertheless, milder winters, cooler summers, and higher

humidity due to the proximity to Lake Michigan may have assisted buffering against geographical gradient effects for decades until the severe drought in 2012 [124].

In the first generation of these five populations, microclimatic effects were utterly overwhelmed by macroclimates. In the second generation of these populations, on the contrary, extensive canopy shading during the summer may have not only serve as refugia for Kbbs to avoid heat stress [89,125–127] but preserve soil moisture to prevent early senescence of wild lupine and other floral resources, elucidating the positive correlations with λ in northwest Wisconsin and Indiana Dunes National Park, both of which were at the species' range limits [128–134]. The negative associations of λ with slope and trasp in the second generation confirmed the findings of [135,136]: in the northern hemisphere, north- and east-facing gentle slopes were much cooler than south- and west-facing steep slopes, leading to higher survivorship of wild lupine and reproductive success of butterflies during the summertime. Nevertheless, poorer model performances of the second than the first generations, according to adjusted-$R^2$ values (Tables 2 and 3), indicated that more complicated interactions with biotic or abiotic environments could be investigated in future studies.

## Conclusions

In a nutshell, the combination of density-dependence, macroclimate, and sometimes microclimate best predicted Kbb population dynamics [79]. Previous studies noted that monthly macroclimate predictors, like mean temperature and total precipitation, exhibited twice the explanatory powers for insect abundance as did extreme weather indicators, perhaps because they had more direct influences on species' physiological responses [137–140]. Although mean temperature was often considered as the primary macroclimatic driver for many pollinator species, our models showed that minimum and maximum temperatures and total precipitation were also vital predictors of Kbbs [93,141,142]. We also ascertain that higher temperatures may generally benefit Kbbs throughout the growing season, while the opposite effect occurred during the wintertime; similarly, for the precipitation, water stress appeared to be a limiting factor for the second generation but not at all for the first generation. As such, our study added new insight that climate change tended to have a mixed effect on this species [93,143–145]. Moreover, significant variables were population-specific, suggesting local adaptation to their isolated habitats. Therefore, integrating populations across regions could obscure patterns of spatial heterogeneity, confirming that models at finer scales could be prioritized in the decision-making process for Kbb conservation.

## Supporting information

**S1 Table. Location details of 48 sampled sites of five Karner blue butterfly (*Lycaeides melissa samuelis*) populations.** The state, county, latitude, and longitude of each site are listed.
(DOCX)

**S1 Dataset. Data inputs for the modeling of first-generation (overwinter) Karner blue butterfly (*Lycaeides melissa samuelis*).** The population name, site, year, densities, macroclimates, and microclimates are included.
(XLSX)

**S2 Dataset. Data inputs for the modeling of second-generation (summer) Karner blue butterfly (*Lycaeides melissa samuelis*).** The population name, site, year, densities, macroclimates, and microclimates are included.
(XLSX)

**S1 File. Sample R codes of statistical analysis and modeling.**
(PDF)

# Acknowledgments

This study was conducted at the University of Minnesota in collaboration with the Institute on the Environment. We thank all the members of the Karner blue butterfly Recovery Team. We especially acknowledge the contributors of Karner blue population count data: Randy Knutson (US National Park Service), Ann Swengel and Scott Swengel (independent researchers), Chelsea Gunther (Wisconsin DNR) and Tim Wilder (Department of Defense, Fort McCoy), Steven Campbell and Neil Gifford (Albany Pine Bush Preserve), and Christopher Hoving (Michigan DNR). Their data are the foundation of this study. We would also like to express our deepest gratitude toward all our reviewers of PLoS One. Their feedback and comments are instrumental, helping us create a much better manuscript.

Any use of trade, product, or firm names is for descriptive purposes only and does not imply endorsement by the U.S. Government.

# Author Contributions

**Conceptualization:** Yudi Li, Ralph Grundel, Jim Perry, Jessica J. Hellmann.

**Data curation:** Yudi Li, Ralph Grundel, Steven Campbell.

**Formal analysis:** Yudi Li, David Wilson.

**Funding acquisition:** Jessica J. Hellmann.

**Investigation:** Yudi Li, Jim Perry, Jessica J. Hellmann.

**Methodology:** Yudi Li, David Wilson, Ralph Grundel, Joseph Knight, Jim Perry, Jessica J. Hellmann.

**Project administration:** Jessica J. Hellmann.

**Software:** Joseph Knight.

**Supervision:** Ralph Grundel, Jessica J. Hellmann.

**Validation:** Yudi Li, David Wilson, Steven Campbell.

**Visualization:** Yudi Li.

**Writing – original draft:** Yudi Li.

**Writing – review & editing:** Yudi Li, David Wilson, Ralph Grundel, Steven Campbell, Joseph Knight, Jim Perry, Jessica J. Hellmann.

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
