## [Decision Letter · Decision Letter 0]

13 Oct 2022

PONE-D-21-39356Extinction risk modeling predicts range-wide differences of climate change impact on Karner blue butterflyPLOS ONE

Dear Dr. Li,

Thank you for submitting your manuscript to PLOS ONE. After careful consideration, we feel that it has merit but does not fully meet PLOS ONE’s publication criteria as it currently stands. Therefore, we invite you to submit a revised version of the manuscript that addresses the points raised during the review process.

We look forward to receiving your revised manuscript.

Kind regards,

Daniel de Paiva Silva, Ph.D.

Academic Editor

PLOS ONE

Journal Requirements:

Additional Editor Comments (if provided):

De Dr. Li,

After this first review round, your manuscript has received a major decision from one of the reviewers and a rejction with potential resubmission from the other reviewer. Consider all the issues raised, and as the reviewer who decided for the rejection allowed the potential resubmission, I decided to provide your manuscript a Major Review. Please note that both reviewers were very criterious and their suggestions should be consdered in full in order to increase your possibilities of acceptance. Please take special attention to the points raised by reviewer #1, but please do not forget to change the text according to what has been proposed by reviewer #2.

Please resubmit your text within a 3-month period. Please provide a rebuttal letter along your text, providing a point-by-point answer to all issues raised by the reviewers. Do not hesitate to resubmit earlier if you can. If you need to solve any doubts with me, do not hesitate to contact me.

Sincerely,

Daniel Silva

Reviewers' comments:

Reviewer's Responses to Questions

**Comments to the Author**

1. Is the manuscript technically sound, and do the data support the conclusions?

Reviewer #1: Partly

Reviewer #2: Partly

2. Has the statistical analysis been performed appropriately and rigorously? 

Reviewer #1: No

Reviewer #2: Yes

3. Have the authors made all data underlying the findings in their manuscript fully available?

Reviewer #1: Yes

Reviewer #2: No

4. Is the manuscript presented in an intelligible fashion and written in standard English?

Reviewer #1: Yes

Reviewer #2: Yes

5. Review Comments to the Author

Reviewer #1: The authors investigate the relationship between abundance measures of Karner blue butterflies and climate variables. Specifically, they look at first and second generation counts of butterflies per kilometer for each year of data across a collection of locations and relate these counts to climate variables as well as previous generation counts, to study their annual variation. In general, I find the paper well-written and the data and application interesting. However, I have some questions pertaining to the model specification and inference that can be obtained. General and specific comments are attached.

Reviewer #2: The study is relevant and brings important questions to be investigated. The choice of this species was quite relevant considering its current status, with good demonstration in the map (Fig 1). However, the introduction does not present hypotheses. I didn't understand why the value of P has p > 0.1 and not p > 0.05 and I would like to know the n value for each sample. I'm not sure if what was done in the present work is really a modeling, without detracting from the merits of the results. Just as we cannot say, from the results found, whether they are really the result of climate change. The tables present the main results and need improve. There are some errors (mentioned below) and I believe that graphs could be added to help reader for better understanding the results found. The choice of variables is also not very clear.

Line 30: For Whole Species the values are negative

Line 39: The insects are not only ectotherms. Many insects are capable of generating endothermic heat. Thus. the sentence is not correct (See Stone and Willmer, 1989 and others)

Lines 40-41: What do you mean with: complex life cycles and varying vulnerability throughout their life spans?

Lines 44-45: this sentence is confusing

Line 47: italics for the species name. Important to add author and year of the species description.

Line 66: (Geiger et al., 2009) need a number

Lines 69-71: the sentence requires reference.

Line 70: how much would be the number of host plants to be considered limited?

LIne 71: The authors don't cite the IUCN Red List of Threatened Species status

Lines 75-77: the sentence requires reference.

Lines 97-101: this sentence is confusing

Lines 119-120: It is necessary to add more detail about sampling. E.g. the size of the transects, hours and days of sample activities

Lines 206-208: Add the statistical reasons found for the statement

Table 1: you put in the legend: Blank cells indicate not significant. However, there are p>0.1 values in the table

Table 2: in the legend only: JN and in the table: JL and JN.

Table 2: PF and in the table: CF. Are the same thing?

Line 243: where are these values?

6. PLOS authors have the option to publish the peer review history of their article (what does this mean?). If published, this will include your full peer review and any attached files.

Reviewer #1: No

Reviewer #2: No

---

## [Author Response · Author response to Decision Letter 0]

12 Jan 2023

Reviewer 1:

 Line 21: "developing adaptive.....?" Missing a word?

Response: Yes, " to develop adaptive management plans " is missing, and has been added back.

Revision: (line 19-20) Evaluating how climate shapes the population trend of the Kbb is necessary to develop adaptive management plans.

 Terminology throughout implies causal inference, yet observational data and regression-based models don't imply causality.

Response: We have changed all the words of causality indicators (e.g., affect, impact, influence, cause, increase/decrease) closely related to the outputs of our models to the alternatives (e.g., associated, predict, likely, risk, probability) in Results and Discussion, except when referring to other experimental studies.

 Line 68: Would survival models be more appropriate in this analysis? How does the survival of generation impact the next?

Response: Survival model is typically used for describing or predicting the survivorship or mortality of a population throughout the individuals' whole lifespan. In our study, the data we have are restricted to sampling on adult Kbbs only, instead of repeated measures on tracked individuals from egg to death. On the other hand, between-generation survival model is rare. The survival rate should range from 0 to 1. However, in our study, though there are decreasing population trends annually, the population size of the second generation is usually larger than the first generation of the same year, resulting in a "survival rate" out of this range. Therefore, we thought that both within-generation and between-generation survival models may not be suitable for our study.

 Line 172: Is this supposed to be Table 2?

Response: Yes, it should be refereeing to S1 Table 2. However, we made two changes in the revised manuscript: 1. S1 Table 2 was moved to the manuscript as Table 1 because this is very important to display more details about the raw data; 2. This sentence was moved from the Statistical Analysis sub-section to the Density-Dependent Data sub-section because we wanted to clarify how we handled this count data earlier (further explained in the response to comment #7).

Revision: (line 102-105) Table 1

 Why fit separate models to the first and second generations? Seems like one model should be fitted with identifiers for the different generations. Indicator variables and interaction terms could be used to assess significant differences between the generations as well as their respective climate drivers.

Response: The two generations occur in different periods of time in a year, mostly nonoverlapped (first generation: from fall & winter of the previous year to June in the current year: second generation: throughout the whole summer--from Jun to Aug). Since we are interested in how the adult density of each generation is affected by the climate patterns occurred within its own lifecycle, including both generations in one model would result in an increase of the degree of freedom due to more unnecessary variables involved. For example, precipitations of Jun, Jul, Aug may have direct impacts on the second generation, but the precipitations of winter and spring do not. Instead, it is the winter and spring precipitations that directly influence the first generation (e.g., survival and reproduction), then in turn affecting the second generation (all individuals are hatched from the eggs of the previous generation), which is also related to the concept of "transgenerational effect" we introduced in the manuscript. Therefore, it's more reasonable for us to include the densities of previous generations into the model rather than the climate drivers in unrelated months that only have indirect effects; otherwise, the number of variables in our models would be tripled. In addition, even if we do not separate models, the interactions between generations and monthly climate variables are not valid and meaningful, because again the second generation never occur in the winter and spring (similar for the first generation in mid-late summer). We tried to better clarify the season why we separated models to avoid confusion in the revised manuscript.

Revision 1: (line 57-62) As a bivoltine species, first-generation Kbbs overwinter as eggs until early April, hatch as larvae feeding on wild lupines and become pupae undergoing a metamorphosis in mid-May and emerge as adults in June with a lifespan of 5-7 days to reproduce. The second generation occurs during the summer, with hatching, pupation, and emergence in early June, late June, and early July, respectively [37]. Their thermal and drought tolerances are specific to each life stage [38-41].

Revision 2: (line 130-132) We modeled the first and the second generation separately because they were mostly non-overlapped except in June and experienced different climate patterns [73-74]. In total, we created 12 models: 6 populations * 2 generations.

 What distributional assumptions are being made by the model? Normal errors? Is this reasonable for this type of data (individuals per km)?

Response: Both the random forest that we stuck to and the partial least square to prevent overfitting and improve interpretability makes no assumptions on the probability density of the response variable. There is no problem to handle different types of data without transformation, including skewed and discrete ones (i.e., density) in our study. Distributional assumptions do matter for the maximum-likelihood-based algorithms such as OLS, GLM, and GAM, but we decided to remove these and other machine learnings that are not random forest or partial least square from the manuscript (further explained in the response to comment #8 below).

Revision: (line 158-159) As a non-parametric approach, both RF and PLS make no assumptions on data distribution shape.

 What exactly are the data for each observation point and how are they aggregated over transects, regions, time? That is, what exactly is being grouped by ecoregion and overall population? How many observations are being used to the model? Just one per year? One per year per ecoregion? One per year per transect within each ecoregion? 

Response 1: The raw data shared with us were the counts of adult Kbb individuals per survey using one transect about 1 km length on each site (or observation point), and several surveys were made per generation (or flight period) per year. We added the total count of individuals into the new Table 1. 

Response 2: The data used for fitting models were densities or derived from densities. We actually made an error on writing out the unit of density: it was individuals per ha instead of per km. Individuals per km is only an intermediate product of our calculation of density. This was corrected in the revised manuscript. 

Response 3: We also decided to remove the concept of ecoregion, because these five populations occupy five different ecoregions. Therefore, it would be more straightforward to make direct inference on each population rather than ecoregion. 

Response 4: We calculated the average density for each site across several transects carried out during a flight period (or generation) per year, so for the models of both generations, the number of "observations" was equal to the number of sites surveyed for a population (new Table 1) in that year (i.e., one per generation per year per site within each population).

Revision 1: (line 94-97) Because the number of surveys per flight period and the total length of transect per survey varied widely across researchers and years, we calculated mean counts per kilometer of transect generation-by-generation for each location. We estimated density as the number of adult individuals per hectare (ha) by assuming that a kilometer of transect represented a 2.5-ha area [64].

Revision 2: (line 128-135) We included both density-dependent and -independent variables to build endo-exogenous models specific to (i) each population, and (ii) all the surveyed locations of the five populations pooled together (All). We modeled the first and the second generation separately because they were mostly non-overlapped except in June and experienced different climate patterns [73-74]. In total, we created 12 models: 6 populations * 2 generations. Each model had λ as the response variable and 18 standardized explanatory variables: two previous-generation densities, four (IT, AT, MT, and PT) * three (egg, larvae/pupae, and flight) monthly climates, three topographies (elevation, slope, and trasp), and tree canopy.

 "Why are the authors interested in the ""six modeling algorithms of R package caret?"" That is, why compare these off-the-shelf models? Is this just exploratory data analysis? If so, remove from the text. If not, justify why it was done and what was gained. Overall, it seems as though the authors want to fit an autoregressive model to study the relationships

between Yt and Xt as well as Yt and Yt-1 (here, Yt is the response at time t, Xt is a set of covariates associated with time t, and Yt-1 is the response at the previous time). If this is true, the authors should write out the model and provide discussion of the utility of each model component."

Response 1: We originally wanted to compare which would work the best for our data, but we just decided to follow your suggestion by removing the process of selecting modeling algorithms from the manuscript and stick to random forest. The SI Table 4 was thus removed. 

Response 2: The model equation was included and explained in the revised manuscript. It's not autoregressive because we used λt instead of Yt (further explained in the response to comment #11 below).

Revision 1: (line 147-152) We applied random forest (RF, “randomForest” R package [77]) for regression because previous studies found that it was more robust to model population trends than regression-based or other machine-learning algorithms [78-79]. We then used the genetic algorithm (GA) of package “caret” for feature selection [80].

Revision 2: (line 141-142) The model equation is expressed as:

λ_t= β_0+β_1 Y_(t-1)+β_2 Y_(t-2)+β_3 〖X1〗_t+⋯+β_17 〖X15〗_t+ε_t

Revision 3: (line 143-146) where λt is density growth rate from the previous generation to the current generation; Yt-1 and Yt-2 is the densities of one and two generations ago, respectively; Xnt are environmental factors of the current generation differing between the two generations. Since there were no “previous generations” for the first-year data, we started the models from the second year.

 Individual model fits (for generations and locations) prevent clear inference in terms of determining significant differences across space and between the two generations. One model that incorporates observations across all locations and generations would be much more useful. That is, stack up the response variables and predictor variables and then specify site specific and generation specific coefficients.

Response 1: We had the models that incorporated all the locations of five populations together - the "All" models. However, this could largely dilute the resolution of our data, and was confirmed by our outputs (Table 2 & Table 3): for instance, it's common to see that the coefficient of a variable was positive in the All model while it became negative in the individual population models. These Kbb populations are highly isolated and may have made local adaptations. Therefore, the differences among populations were what we expected and having models at population scale could help us better capture the divergences. We actually used the All models as our reference (or "broad picture") to prove this.

Response 2: For the generation, as we explained in the response to comment #5, we still don't think combining two generations into one model is in the right direction. Again, these two generations are influenced by different factors (both density-dependent and -independent variables) because they are not occurring at the same time in a year and the second-generation density depends on the first generation, and thus they could not be easily stacked up. This is actually the core of this study and is also the reason why we called this as a "novel approach".

Revision: (line 128-131) We included both density-dependent and -independent variables to build endo-exogenous models specific to (i) each population, and (ii) all the surveyed locations of the five populations pooled together (All). We modeled the first and the second generation separately because they were mostly non-overlapped except in June and experienced different climate patterns [73-74].

 line 268: How do you assess which is more influential?

Response: We thought you were asking about line 258. We used the estimated coefficient of PLS models to make this statement. The variable importance analysis on RF models also showed greater influences of density dependence on population growth rate, though they were not shown in the manuscript because it was highly consistent with PLS outputs.

Revision: (line 214-215) However, the density of the second generation in the previous year (PS) almost always had larger regression coefficients than the other density-dependent variable.

 How does a model that regresses Yt on Yt-1 lead to inference with regard to density dependence? Density dependence typically refers to a relationship between population growth rate and population density. If I'm understanding the model as written, you are actually just identifying positive autocorrelation.

Response: We originally have models for both Yt and λt. The model performances of Yt were generally better than λt, so we included Yt in the previous manuscript. However, we realized that using Yt did not confer the conventional definition of density dependence and could rise the question on autocorrelation, we decided to switch to the models on λt.

Revision 1: (line 100-101) We also calculated the population growth rate (λ) based on density change between current and previous generations.

Revision 2: (line 132-135) Each model had λ as the response variable and 18 standardized explanatory variables: two previous-generation densities, four (IT, AT, MT, and PT) * three (egg, larvae/pupae, and flight) monthly climates, three topographies (elevation, slope, and trasp), and tree canopy.

 What is the model for Y1, i.e., how do you model the count data from year 1?

Response: We did not model year 1; we started from year 2.

Revision: (line 145-146) Since there were no “previous generations” for the first-year data, we started the models from the second year.

Reviewer 2:

 Line 30: For Whole Species the values are negative.

Response: Yes, in the previous manuscript, Whole Species values were negative, which was the reason why we mentioned "generally". In the revised manuscript, since we decided to stick to modeling the growth rate of density instead of the density itself, the outputs were different from the previous version (see the Response to Comment #11 of Reviewer 1 for more details).

Revision: (line 29-31) The growth rate decreased with higher overwinter temperatures and spring precipitation in the first generation, while it increased with higher summer temperatures and precipitations in the second generation.

 Line 39: The insects are not only ectotherms. Many insects are capable of generating endothermic heat. Thus. the sentence is not correct (See Stone and Willmer, 1989 and others).

Response: Yes, this is our mistake. We decided to remove the concept of ectotherms and focus on butterflies instead of insects in general.

Revision: (line 35-37) Butterflies, as good indicator species with rapid responses to fluctuations of abiotic conditions [1-6], are highly sensitive and vulnerable to climate change because many have reached the maximum adaptive capacity [7-8].

 Lines 40-41: What do you mean with: complex life cycles and varying vulnerability throughout their life spans?

Response: This sentence could be confusing. Basically, different life stages in the life cycles of insects (now butterflies) have unique interactions with climate. We have made our expression clearer in the revised manuscript.

Revision: (line 37-38) Climate change interferes with butterflies by impacting development throughout the life cycle and changing interaction with host plants.

 Lines 44-45: this sentence is confusing.

Response: The whole paragraph is more relevant to another manuscript of us where we used the models created in this manuscript to predict extinction risk and distribution on future climate change for Kbbs. So we decided to remove the whole paragraph.

Revision: (previously line 43-49) Removal of the whole paragraph.

 Line 47: italics for the species name. Important to add author and year of the species description.

Response: The scientific name was changed to italic form. The author and the year of description were also added.

Revision: (line 45-46) The Karner blue butterfly (Lycaeides melissa samuelis, or Kbb), first described by Vladimir Nabokov in 1944, was listed as federally endangered in 1992 [23-25].

 Line 66: (Geiger et al., 2009) need a number.

Response: The reference number should be ‘55’ and we forgot to convert it into this number. We also improved the sentence a little bit.

Revision: (line 66-68) For instance, south-facing slopes are generally warmer than north-facing slopes in the northern hemisphere; higher elevations are colder and drier than lower altitudes [55-56].

 Lines 69-71: the sentence requires reference.

Response: We added back the reference numbers at the end of this sentence.

Revision: (line 46-51) Like many species in the family Lycaenidae, it is vulnerable to climate change because of limited dispersal ability (< 1 km), single host plant (i.e., monophagy on wild lupine, Lupinus perennis), and habitat specialist relying on high-quality oak savanna and pine barren which have been lost and fragmented for decades as the result of fire suppression, agricultural intensification, and urbanization [26-30].

 Line 70: how much would be the number of host plants to be considered limited?

Response: we tried to make it clearer by using the phrase "monophagy" and we explicitly state "single host plant" in the revised sentence.

Revision: (line 46-51) Like many species in the family Lycaenidae, it is vulnerable to climate change because of limited dispersal ability (< 1 km), single host plant (i.e., monophagy on wild lupine, Lupinus perennis), and habitat specialist on high-quality oak savanna and pine barren which have been lost and fragmented for decades as the result of fire suppression, agricultural intensification, and urbanization [26-30].

 Line 71: The authors don't cite the IUCN Red List of Threatened Species status.

Response: Has been added, and the reference number is 23.

Revision: (line 45-46) The Karner blue butterfly (Lycaeides melissa samuelis, or Kbb), first described by Vladimir Nabokov in 1944, was listed as federally endangered in 1992 [23-25].

 Lines 75-77: the sentence requires reference.

Response: The sentence was incorporated into the same one responding to Comments #7 and #8.

Revision: (line 46-51) Like many species in the family Lycaenidae, it is vulnerable to climate change because of limited dispersal ability (< 1 km), single host plant (i.e., monophagy on wild lupine, Lupinus perennis), and habitat specialist on high-quality oak savanna and pine barren which have been lost and fragmented for decades as the result of fire suppression, agricultural intensification, and urbanization [26-30].

 Lines 97-101: this sentence is confusing.

Response: The sentence was modified and embedded first paragraph.

Revision: (line 40-44) However, the impacts are mixed: it was found that warmer temperature facilitates egg and pupa survival, advances flight period, alters female fertility, and induces extra generations [13-15], while it decreases larva survival and adult body size [16-18]. Drought can reduce food availability and delay growth, but excessive rainfall may disrupt nectar and pollen foraging as well as egg laying [19-22].

 Lines 119-120: It is necessary to add more detail about sampling. E.g. the size of the transects, hours and days of sample activities.

Response: More details about the sampling have been added to the text and new Table 1 (previously in SI).

Revision 1: (line 86-93) We used time-series data, of 8–27-year duration, from five Kbb populations across the species’ range (Table 1). Four populations were located on oak savannas in the Midwest U.S.: central Wisconsin (CW), northwest Wisconsin (NW), Fort McCoy (FM), and Indiana Dunes National Park (IDNP). One was from pine barrens in the eastern U.S.—Albany Pine Bush Preserve (APBP). Our data represent adult Kbbs surveyed at multiple locations (S1 Table) along one to several linear transects per location repeatedly every 7-14 day interval during the two flight periods (i.e., from late May to late June for the first generation, and from mid-July to mid-August for the second generation).

Revision 2: (line 102-105) Table 1.

 Lines 206-208: Add the statistical reasons found for the statement.

Response: We added the p-value into the statement, though the negative associations were very straightforward based on Table 2 (previously Table 1).

Revision: (line 173-175) For the first generation, λ was negatively correlated with PS for All, CW, NW, and FM populations (p < 0.05) and with PF for IDNP and APB populations (p < 0.01, Table 2).

 Table 1: you put in the legend: Blank cells indicate not significant. However, there are p>0.1 values in the table.

Response: In the new Table 2 (previously Table 1), we added in the legend b that "ns" indicates nonsignificant, and in the lengend c that blank cells represented exclusion from the best-fitted models.

Revision: (line 184-185) Table 2 legend b and c.

 Table 2: in the legend only: JN and in the table: JL and JN.

Response: The explanation of JL has been added to legend e at the bottom of new Table 3 (previously Table 2).

Revision: (line 208-210) Table 3 legend e.

 Table 2: PF and in the table 1: CF. Are the same thing?

Response: No, PF is the density of previous-year first generation and CF is the density of previous-year second generation. We revised the legends at the bottom of new Table 3.

Revision: (line 208-210) Table 3 legend e.

 Line 243: where are these values?

Response: The adjusted-R2 are at the last row of the table. We indicated them in the title of Table 2 and 3.

Revision: (line 182-183 & line 202-203) The last row of Table 2 and 3.

Editor

Response: We have modified all the documents to comply with the requirements.

 In your Methods section, please provide additional information regarding the permits you obtained for the work. Please ensure you have included the full name of the authority that approved the field site access and, if no permits were required, a brief statement explaining why.

Response: The authority names were included.

 We note that Figure 1 in your submission contain [map/satellite] images which may be copyrighted. All PLOS content is published under the Creative Commons Attribution License (CC BY 4.0), which means that the manuscript, images, and Supporting Information files will be freely available online, and any third party is permitted to access, download, copy, distribute, and use these materials in any way, even commercially, with proper attribution. For these reasons, we cannot publish previously copyrighted maps or satellite images created using proprietary data, such as Google software (Google Maps, Street View, and Earth). For more information, see our copyright guidelines: http://journals.plos.org/plosone/s/licenses-and-copyright. We require you to either (1) present written permission from the copyright holder to publish these figures specifically under the CC BY 4.0 license, or (2) remove the figures from your submission.

Response: We decided to remove Fig 1.

---

## [Decision Letter · Decision Letter 1]

9 Mar 2023

PONE-D-21-39356R1Extinction risk modeling predicts range-wide differences of climate change impact on Karner blue butterflyPLOS ONE

Dear Dr. Li,

Thank you for submitting your manuscript to PLOS ONE. After careful consideration, we feel that it has merit but does not fully meet PLOS ONE’s publication criteria as it currently stands. Therefore, we invite you to submit a revised version of the manuscript that addresses the points raised during the review process.

We look forward to receiving your revised manuscript.

Kind regards,

Daniel de Paiva Silva, Ph.D.

Academic Editor

PLOS ONE

Journal Requirements:

Additional Editor Comments:

Dear Dr. Li,

After this new review round, both reviewers believe your manuscript is nearly suitable for publication in PLoS One. Consequently, both of them provided a minor review status for you manuscript. Please proceed with the changes suggested by the reviewers and I believe your manuscript will be accepted for publication.

Sincerely,

Daniel Silva

Reviewers' comments:

Reviewer's Responses to Questions

**Comments to the Author**

1. If the authors have adequately addressed your comments raised in a previous round of review and you feel that this manuscript is now acceptable for publication, you may indicate that here to bypass the “Comments to the Author” section, enter your conflict of interest statement in the “Confidential to Editor” section, and submit your "Accept" recommendation.

Reviewer #2: (No Response)

Reviewer #3: (No Response)

2. Is the manuscript technically sound, and do the data support the conclusions?

Reviewer #2: Yes

Reviewer #3: Yes

3. Has the statistical analysis been performed appropriately and rigorously? 

Reviewer #2: Yes

Reviewer #3: Yes

4. Have the authors made all data underlying the findings in their manuscript fully available?

Reviewer #2: Yes

Reviewer #3: Yes

5. Is the manuscript presented in an intelligible fashion and written in standard English?

Reviewer #2: Yes

Reviewer #3: Yes

6. Review Comments to the Author

Reviewer #2: Line 208 you put: CF: density of current-year first-generation, no like density of previous-year second generation as you mentioned

Reviewer #3: Comments and reviewer recommendation for manuscript number PONE-D-21-39356R1 “Extinction risk modelling predicts range-wide differences of climate change impact on Karner blue butterfly” by Li et al.

This manuscript deals with an interesting topic as it tries to model the extinction risk of an endangered butterfly in the US. Since it is a revision of a previously submitted manuscript, I find the replies to the reviewers very thorough and correct. They clearly improved the manuscript and it would be nice to thank them for that in the Acknowledgements.

The analyses in this manuscript are sound and are well-presented in a table. In the text, however, they are not! The results section (lines 173-199) only lists the correlations that are significant and uses only abbreviations that are totally unreadable for people that are unfamiliar with this work. This is OK in a table, but not in a text. Here, the results should be described ecologically, so not just saying that X is higher than Y or the correlation between X and Y is significant, but explaining what these outcomes mean in plain and understandable text such as “population density increased in the second generation with increased summer temperatures in most populations, but increased rainfall in August resulted in lower population densities in the first generation especially in populations at the northern edge of their range (THIS IS A FICTITIOUS EXAMPLE!).

Apart from this comment, I found this manuscript well-written with adequate references (I only suggested one additional possible reference in the attached annotated pdf). This paper is a valuable contribution to the knowledge of how climate change impacts on threatened species!

More detailed remarks are given in het attached pdf.

7. PLOS authors have the option to publish the peer review history of their article (what does this mean?). If published, this will include your full peer review and any attached files.

Reviewer #2: No

Reviewer #3: No

---

## [Author Response · Author response to Decision Letter 1]

22 Jul 2023

Reviewer 2:

1. Line 208 you put: CF: density of current-year first-generation, no like density of previous-year second generation as you mentioned

Response: We changed the description of “CF” to be consistent with the others.

Revision: (line 194) PS: density of previous-year second generation; PF: density of previous-year first generation; …

Reviewer 3:

1. Since it is a revision of a previously submitted manuscript, I find the replies to the reviewers very thorough and correct. They clearly improved the manuscript and it would be nice to thank them for that in the Acknowledgements.

Response: We added a statement of acknowledgement to the reviewers.

Revision: (line 308-310) We would also like to express our deepest gratitude toward all our reviewers of PLoS One. Their feedback and comments are instrumental, helping us create a much better manuscript.

2. The results section (lines 173-199) only lists the correlations that are significant and uses only abbreviations that are totally unreadable for people that are unfamiliar with this work. This is OK in a table, but not in a text. Here, the results should be described ecologically, so not just saying that X is higher than Y or the correlation between X and Y is significant, but explaining what these outcomes mean in plain and understandable text such as “population density increased in the second generation with increased summer temperatures in most populations, but increased rainfall in August resulted in lower population densities in the first generation especially in populations at the northern edge of their range (THIS IS A FICTITIOUS EXAMPLE!).

Response: We strongly agreed and realized this indeed could cause a lot of confusion. We thus completely revised the two paragraphs in the Results, following the fictitious example provided, to make them more readable and easier to understand and interpret.

Revision: (Paragraph 1; line 173-186) In the first generation (Table 2), λ decreased with increased density of previous-year second generation for the All model and the three populations in Wisconsin (CW, NW, and FM) with p-values < 0.05. However, for IDNP and APBP populations, it was the density of previous-year first generation that displayed the significant inverse correlation with λ (p < 0.01). In more than one population across their distributional range, λ was consistently lower with higher overwinter temperatures (mean, max, and min), spring total precipitation and June min temperature, though these negative relationships were not always significant. Overwinter precipitation and spring max temperature were only significant to APBP (p < 0.001 and p < 0.01, respectively), the population not in the Upper Midwest. The rest macroclimate variables, including spring min temperature, June mean and max temperatures, and June total precipitation were not contained in any best-fitted model. None of the macroclimatic variables were selected by the GA approach in the model of NW, the population at the northern edge of Kbb’s range. Although all the four topo-climatic variables were left in the final models of NW and IDNP, they were non-significant (p > 0.05) in both populations.

Revision: (Paragraph 2; line 198-208) In the second generation (Table 3), λ increased with higher density of previous-year second generation, June min temperature, July total precipitation, and August mean, min, and max temperatures in more than one population with p-values < 0.05). It also uniquely increased with higher June mean temperature and August total precipitation (p < 0.05) in CW and NW, respectively. The density of current-year first generation was non-significant, and June max temperature and total precipitation were absent in all models. . Unlike the first generation, tree canopy and slope became significant in multiple populations with consistent positive and negative correlations, respectively, and in particular, the p-values of slope were smaller than 0.01 for FM and IDNP, both of which were located closer to the southern edge of Kbb’s range than the other three populations. For INDP specifically, λ further decreased with larger trasp, and none of the density-dependent and macroclimatic variables were significant.

3. I only suggested one additional possible reference in the attached annotated pdf.

Response: Unfortunately, we did not find the attached annotated pdf in the email and on the editorial manager. The only available pdf is the one containing the comments of the Reviewer 1 during the last round of revision.

---

## [Decision Letter · Decision Letter 2]

30 Aug 2023

PONE-D-21-39356R2Extinction risk modeling predicts range-wide differences of climate change impact on Karner blue butterflyPLOS ONE

Dear Dr. Li,

Thank you for submitting your manuscript to PLOS ONE. After careful consideration, we feel that it has merit but does not fully meet PLOS ONE’s publication criteria as it currently stands. Therefore, we invite you to submit a revised version of the manuscript that addresses the points raised during the review process.

We look forward to receiving your revised manuscript.

Kind regards,

Daniel de Paiva Silva, Ph.D.

Academic Editor

PLOS ONE

Journal Requirements:

Additional Editor Comments:

Dear Dr. Li,

After this new review round, the remaining reviewer believes the MS is almost there. Still, they observed that there are several abbreviations that still need to be resolved and some minor other issues that need to be corrected before the manuscript is accepted for publication. As soon as the MS is corrected, it will be re-evaluated and if everything checks out, it will be accepted for publication.

Sincerely,

Daniel Silva.

Reviewers' comments:

Reviewer's Responses to Questions

**Comments to the Author**

1. If the authors have adequately addressed your comments raised in a previous round of review and you feel that this manuscript is now acceptable for publication, you may indicate that here to bypass the “Comments to the Author” section, enter your conflict of interest statement in the “Confidential to Editor” section, and submit your "Accept" recommendation.

Reviewer #3: All comments have been addressed

2. Is the manuscript technically sound, and do the data support the conclusions?

Reviewer #3: Yes

3. Has the statistical analysis been performed appropriately and rigorously? 

Reviewer #3: Yes

4. Have the authors made all data underlying the findings in their manuscript fully available?

Reviewer #3: No

5. Is the manuscript presented in an intelligible fashion and written in standard English?

Reviewer #3: Yes

6. Review Comments to the Author

Reviewer #3: Comments and reviewer recommendation for manuscript number PONE-D-21-39356R2 “Extinction risk modelling predicts range-wide differences of climate change impact on Karner blue butterfly” by Li et al.

This revised version is ok, apart from one thing I apparently did not notice during the previous readings of the manuscript. When reading it now, I find that there are a lot of abbreviations that might be familiar to the authors, but not to an international readership. I give a (somewhat exaggerated and fictious) example to make my case: “The λ of the Kbb in the IDNP and APBP populations in the NW was mostly determined by PF and not by AT, MT or PT resulting form the GA approach” … :-D As a reviewer I have to read the whole manuscript and I try to remember as many abbreviations as possible (NOT!), but as a scientist/reader, I usually only read parts of a paper (last paragraph of the introduction, conclusion, figures …). When I come across sentences like the one I wrote, I usually stop reading because I have to struggle to hard to find out what all the abbreviations mean and that would be a pity for this very interesting work! A simple figure with a map where the study sites are located, together with the abbreviations if you want, would make localising the sites at least a lot easier (not everyone is American and knows where all American states are situated). In the era of printed versions, I can understand that limiting the number of characters was important, but with (almost) everything being digital, the use of full words instead of abbreviations would make it much more reader-friendly. If the editor of PONE is ok with abbreviations, fine by me, but I don’t like it …

The potential additional reference that could be included in the section about climate warming leading to extra generations and the possible negative consequences of it is “Van Dyck H, Puls R, Bonte D, Gotthard K & Maes D (2015) The lost generation hypothesis: could climate change drive ectotherms into a developmental trap? Oikos 124 (1): 54-61. https://doi.org/10.1111/oik.02066.” but do not feel obliged to include it by any means.

7. PLOS authors have the option to publish the peer review history of their article (what does this mean?). If published, this will include your full peer review and any attached files.

Reviewer #3: No

---

## [Author Response · Author response to Decision Letter 2]

29 Sep 2023

Reviewer 3:

1. This revised version is ok, apart from one thing I apparently did not notice during the previous readings of the manuscript. When reading it now, I find that there are a lot of abbreviations that might be familiar to the authors, but not to an international readership. I give a (somewhat exaggerated and fictious) example to make my case: “The λ of the Kbb in the IDNP and APBP populations in the NW was mostly determined by PF and not by AT, MT or PT resulting form the GA approach” … :-D As a reviewer I have to read the whole manuscript and I try to remember as many abbreviations as possible (NOT!), but as a scientist/reader, I usually only read parts of a paper (last paragraph of the introduction, conclusion, figures …). When I come across sentences like the one I wrote, I usually stop reading because I have to struggle to hard to find out what all the abbreviations mean and that would be a pity for this very interesting work! In the era of printed versions, I can understand that limiting the number of characters was important, but with (almost) everything being digital, the use of full words instead of abbreviations would make it much more reader-friendly. If the editor of PONE is ok with abbreviations, fine by me, but I don’t like it …

Response: We went through the Methods, Results, and Discussion sections and changed all the abbreviations that may cause confusions.

Revision: For instance, (line 176 – 177) the two abbreviated population names IDNP and APBP were revised to Indiana Dunes National Park and Albany Pine Bush Preserve for clarification.

2. A simple figure with a map where the study sites are located, together with the abbreviations if you want, would make localising the sites at least a lot easier (not everyone is American and knows where all American states are situated). 

Response: We added a muti-paneled figure mapping out the locations of all the permanent sampled sites of the five populations of Karner blue butterfly.

Revision: (line 93 – 97) Captions of Fig 1.

3. The potential additional reference that could be included in the section about climate warming leading to extra generations and the possible negative consequences of it is “Van Dyck H, Puls R, Bonte D, Gotthard K & Maes D (2015) The lost generation hypothesis: could climate change drive ectotherms into a developmental trap? Oikos 124 (1): 54-61. https://doi.org/10.1111/oik.02066.” but do not feel obliged to include it by any means.

Response: We thought this potential refence was relevant to our manuscript, as we did briefly mention the potential of a third generation. We therefore still decided to add this reference and write out a new sentence about the caveat due to the “developmental trap” demonstrated by Van Dyck et al. (2015).

Revision: (line 257 – 258) However, the envisaged third generation, if present, worth further research as this could be a maladaptive response if novel conditions in fall were unfavorable [111].

---

## [Editor Report · Decision Letter 3]

3 Oct 2023

Extinction risk modeling predicts range-wide differences of climate change impact on Karner blue butterfly

PONE-D-21-39356R3

Dear Dr. Li,

We’re pleased to inform you that your manuscript has been judged scientifically suitable for publication and will be formally accepted for publication once it meets all outstanding technical requirements.

Kind regards,

Daniel de Paiva Silva, Ph.D.

Academic Editor

PLOS ONE
---

## [Editor Report · Acceptance letter]

27 Oct 2023

PONE-D-21-39356R3 

Extinction risk modeling predicts range-wide differences of climate change impact on Karner blue butterfly (*Lycaeides melissa samuelis*) 

Dear Dr. Li:

I'm pleased to inform you that your manuscript has been deemed suitable for publication in PLOS ONE. Congratulations! Your manuscript is now with our production department. 

Kind regards, 

on behalf of

Dr. Daniel de Paiva Silva 

Academic Editor

PLOS ONE